# Entanglement of Pseudo-Hermitian Random States

**DOI:** 10.3390/e22101109

**Published:** 2020-09-30

**Authors:** Cleverson Andrade Goulart, Mauricio Porto Pato

**Affiliations:** Instituto de Física, Universidade de São Paulo, Caixa Postal 66318, 05314-970 São Paulo, Brazil; cleversonagoulart@gmail.com

**Keywords:** quantum information, von Neumann entropy, random matrix theory, pseudo-Hermitian operators, PT-symmetry

## Abstract

In a recent paper (A. Fring and T. Frith, Phys. Rev A **100**, 101102 (2019)), a Dyson scheme to deal with density matrix of non-Hermitian Hamiltonians has been used to investigate the entanglement of states of a *PT*-symmetric bosonic system. They found that von Neumann entropy can show a different behavior in the broken and unbroken regime. We show that their results can be recast in terms of an abstract model of pseudo-Hermitian random matrices. It is found however that although the formalism is practically the same, the entanglement is not of Fock states but of Bell states.

## 1. Introduction

The importance of the authors’ findings of Ref. [1] lies in the consequences it has to the question of time evolution of non-Hermitian Hamiltonians [2,3], or, more precisely, to the evolution of their associated density matrices. It shows that by introducing an appropriate time-dependent metric, the density matrix of a non-Hermitian Hamiltonian can be linked to one of a Hermitian one by a similarity transformation. Therefore, both share the same von Neumamm entropy. From a practical point of view, Dyson’s formula is an equation of motion that connects the metric and the Hamiltonians that in principle, can be solved as shown in [1]. On the other hand, the identification in Ref. [1] of a regime in which entanglement is preserved in the time evolution is an important result with potential application in quantum computation.

The non-Hermiticity addressed in Refs. [1,4,5] is that related to the *PT*-symmetry. It can be shown that a complex non-Hermitian Hamiltonian invariant under the combined parity (*P*) and time reversal (*T*) transformations have eigenvalues which are real or complex conjugate. Assuming that an operator is connected to its adjoint by a similarity transformation
(1)$$A^{†}=\eta A\eta^{-1},$$
in which η is a Hermitian operator, then its eigenvalues are real or complex conjugate. Operators satisfying this condition have been defined as belonging to the class of pseudo-Hermitian operators [6,7,8]. This follows from the fact that using the operator η as a metric, the internal product can be redefined such that quantum mechanics relations can be extended to the case of PT symmetric Hamiltonians [9,10,11,12].

Since the beginning of the studies of PT-symmetric systems there was an interest in investigating random matrix ensemble to model properties of this kind of Hamiltonians. This comes naturally as time reversal symmetry plays an important role in RMT. Several ensembles already have been proposed [13,14,15,16] but here we focus on the recently introduced ensemble of pseudo-Hermitian random matrices [17,18,19,20]. However, the matrices in these references have elements which are Gaussian distributed while constructing our model, naturally we were lead to the so-called Wishart matrices in which elements are not Gaussian distributed. In the context of the application of random matrix ensemble to entanglement, the Wishart ensemble has been used to model the Schmidt eigenvalues [21,22,23] of bipartite systems. In our model, however, the Wishart ensemble enters in a different way as is described below.

Another aspect of the model that we want to highlight is that its structure is analogous to that of the well-established quantum gates [24]. In light of the recent interest in the application of random matrices in entanglement phenomena [25,26,27], the task of finding Bell state [28,29] equivalents for the matrices of this ensemble and studying their time evolution provides valuable insight into the physical nature and potential applications of such ensembles.

Organizing the article, we decided to put the description of Dyson’s scheme in the Appendix A. In Section 2, the four random matrices which are the basic ingredients of the model are introduced and their properties are discussed. In Section 3, two pseudo-Hermitian Hamiltonians are defined, their time evolutions are derived and the two regimes of the entanglement of their states are obtained. Finally, in the conclusion section, the physical implications of the results are discussed.

## 2. Pauli-Like Random Matrices

Using as tools projector operators *P* and *Q* defined as [30]
(2)$$P=\sum_{i=1}^{M}|i\rangle \langle i|\;\;\mathrm{and}\;\;Q=\sum_{j=M+1}^{N}|j\rangle \langle j|$$
such that P+Q=1, a rectangular block of dimension M×(N−M) with N≥2M can be singled out from a Gaussian matrix *H* of the Random Matrix Theory to construct with it, the N×N matrix
(3)$$W=PHQ=\left(\begin{array}{cc} 0_{M\times M} & H_{M\times N-M}\\ 0_{N-M\times M} & 0_{N-M\times N-M} \end{array}\right),$$
where $0_{a\times b}$ is an order a×b rectangular matrix containing all zeros, and $H_{a\times b}$ is the order a×b rectangular block from the matrix *H*. Combining then *W* with its adjoint, $W^{†}=QHP$, three new Hermitian traceless matrices R,S and *T* can be defined as
(4)$$R=W+W^{†},\;\;S=-i\left(W-W^{†}\right),\;\;\mathrm{and}\;\;T=WW^{†}-W^{†}W.$$Furthermore, by adding to these matrices, the matrix
(5)$$U=WW^{†}+W^{†}W,$$
it is found that the commutation relations
(6)[R,S]=2iT,
(7)[S,T]=2iRU
and
(8)[T,R]=2iSU.
are satisfied. As *U* commutes with the others and $R^{2}+S^{2}+T^{2}=2U+U^{2},$ we can say that it is a generalized Casimir operator. It is notable that these matrices have the structure of typical quantum gates [24,29]. Specifically, *R*, *S* and *T* have the structure of Pauli −X, −Y and −Z gates, respectively.

As *H* have Gaussian elements, $WW^{†}$ and $W^{†}W$ are Wishart–Laguerre matrices of sizes M×M and (N−M)×(N−M), respectively. Consequently, the eigenstates of *U* are made of the decoupled eigenstates of its separated blocks. Let us denote by $|x_{k}\rangle$ with k=1,2,…,M and $|y_{l}\rangle$ with l=1,2,…,N−M the separated orthonormal eigenstates of this bipartite system; specifically, we have
(9)$$U|x_{k}\rangle =WW^{†}|x_{k}\rangle =x_{k}|x_{k}\rangle$$
and
(10)$$U|y_{l}\rangle =W^{†}W|y_{l}\rangle =y_{l}|y_{l}\rangle$$
with $\langle y_{l}|x_{k}\rangle =0$.

As a matter of fact, more can be said about these states. Indeed, by multiplying the eigenvalue equation of $|x_{k}\rangle$ by $W^{†},$ that is
(11)$$W^{†}\left(WW^{†}\right)|x_{k}\rangle =\left(W^{†}W\right)W^{†}|x_{k}\rangle =x_{k}W^{†}|x_{k}\rangle$$
we find that $W^{†}|x_{k}>$ is an eigenvector of $W^{†}W$ with the same eigenvalue. Moreover, as $\mathrm{tr}\left(U\right)=\mathrm{tr}\left(WW^{†}\right)+\mathrm{tr}\left(W^{†}W\right)=2\mathrm{tr}\left(WW^{†}\right),$ the fact that the two operators share the same set of *M* eigenvalues implies that all the other eigenvalues of $W^{†}W$ are zero. For simplicity, we shall concern ourselves with the case in which M≤N/2, but the results are easily extendable for the M>N/2 and quantitatively analogous. We can, therefore, assume the first *M* normalized eigenvectors of the operator $W^{†}W$ to be given by
(12)$$|y_{l}\rangle =\frac{W^{†}}{\sqrt{x_{l}}}|x_{l}\rangle \;\mathrm{with}\;l=1,\ldots ,M$$

Taking into account the above bases, we consider diagonalized and reduced operators given by
(13)$$\hat{U}=\sum_{k=1}^{M}|x_{k}\rangle x_{k}\langle x_{k}|+\sum_{l=1}^{M}|y_{l}\rangle x_{l}\langle y_{l}|$$
(14)$$\hat{R}=\sum_{k=1}^{M}|x_{k}\rangle \sqrt{x_{k}}\langle y_{k}|+\sum_{l=1}^{M}|y_{l}\rangle \sqrt{y_{l}}\langle x_{l}|$$
(15)$$\hat{S}=-i\sum_{k=1}^{M}|x_{k}\rangle x_{k}\langle y_{k}|+i\sum_{l=1}^{M}|y_{l}\rangle x_{l}\langle x_{l}|$$
(16)$$\hat{T}=\sum_{k=1}^{M}|x_{k}\rangle x_{k}\langle x_{k}|-\sum_{l=1}^{M}|y_{l}\rangle x_{l}\langle y_{l}|$$
that act in the space of dimension $M^{2}$. We remark that this result is consistent with the Schmidt decomposition of a bipartite system [31]. Accordingly, $\hat{R},$
$\hat{S}$ and $\hat{T}$ have eigenvalues $\pm \sqrt{x_{k}}$ with normalized eigenvectors
(17)$$|X_{k}^{\pm }\rangle =\frac{1}{\sqrt{2}}\left(|x_{k}\rangle \pm |y_{k}\rangle \right),$$
(18)$$|Y_{k}^{\pm }\rangle =\frac{1}{\sqrt{2}}\left(|x_{k}\rangle \pm i|y_{k}\rangle \right)$$
and
(19)$$|Z_{k}^{+}\rangle =|x_{k}\rangle \;\mathrm{and}\;|Z_{k}^{-}\rangle =|y_{k}\rangle$$
respectively.

This also implies that those interactions may be seen as analogues to the effect of quantum gates on qu-*d*it states, where the digit is determined by *M* [32]. Specifically, the corresponding Bell state is composed of two qu-*M*its, corresponding to the dimension of the images of $WW^{†}$ and $W^{†}W$.

Moreover, as $\hat{U}$ is positive, it can be used to perform the polar decomposition [31] of our matrices such that a vector operator can be defined as $g=({\hat{U}}^{-1/2}\hat{R},{\hat{U}}^{-1/2}\hat{S},{\hat{U}}^{-1}\hat{T})$. In terms of the components of the operator g, the above commutation relations can be expressed as
(20)$$\left[g_{i},g_{j}\right]=2i\varepsilon_{ijk}g_{k},$$
which together with the anti-commutation relations
(21)$$\left\{g_{i},g_{j}\right\}=2\delta_{ij}1,$$
where 1 is the identity matrix, confirm that the set of unitary matrices $g_{i}$ have the same structure of the Pauli matrices. They are therefore a SU(2) representation. In fact, for M=1 and N=2, the three $g_{i}$ matrices indeed coincide with the Pauli matrices. For M>1, the $g_{i}$ are extensions of the Pauli matrices in which the zeros and the ones have dimension *M*, namely they are $0_{M\times M}$ and $1_{M\times M}$. We also notice that the Baker–Campbell–Hausdorff formula provides the useful expansion
(22)$$\exp\left(ag_{i}\right)g_{j}\exp\left(-ag_{i}\right)=g_{j}\cosh\left(2a\right)+\frac{1}{2}\left[g_{i},g_{j}\right]\sinh\left(2a\right)\;\mathrm{with}\;i\neq j.$$

Using the Bloch vector u=(sinθcosϕ,sinθsinϕ,cosθ), the operator
(23)$$\begin{array}{cc} \frac{1+u\cdot g}{2} & ={\hat{U}}^{-1}\left(\frac{\hat{U}+\hat{T}}{2}\cos^{2}\frac{\theta }{2}+\frac{\hat{U}-\hat{T}}{2}\sin^{2}\frac{\theta }{2}\right)\\ \; & +{\sqrt{\hat{U}}}^{-1}\left(\frac{\hat{R}+i\hat{S}}{2}\frac{\sin\theta }{2}e^{-i\phi }+\frac{\hat{R}-i\hat{S}}{2}\frac{\sin\theta }{2}e^{+i\phi }\right) \end{array}$$
is constructed that has eigenvalue one and eigenvectors
(24)$$\cos\frac{\theta }{2}|x_{k}\rangle +\sin\frac{\theta }{2}e^{i\phi }|y_{k}\rangle$$
and
(25)$$\sin\frac{\theta }{2}|y_{l}\rangle +\cos\frac{\theta }{2}e^{-i\phi }|x_{l}\rangle .$$
Therefore, it is a projector and the eigenvectors are pure states in the Bloch sphere.

The above operator is a Hermitian linear combination of the generators. By making imaginary one of the coefficients, non-Hermitian linear combination of these matrices can be introduced. Once this is done, Dyson’s formula can then be used to eliminate the non-Hermitian term in such a way that an associated Hermitian Hamiltonian is produced. In the next section, this procedure is applied to two non-Hermitian Hamiltonians.

## 3. The Pseudo-Hermitian Hamiltonians

By linearly superposing the operators $\hat{U},\hat{R},\hat{S}$ and $\hat{U},\hat{T},\hat{S},$ two Hamiltonians are constructed in which the matrix $\hat{S}$ plays the role of the non-Hermitian term. We show how, in both cases, the time evolution leads to entanglement of, respectively, the chiral states of $\hat{R}$ in the first Hamiltonian and the bipartite states of $\hat{T}$ in the second one.

### 3.1. Entanglement of Chiral States

We start considering the case of the Hamiltonian
(26)$$A_{1}=\hat{U}+b\hat{R}+ic\hat{S}$$
where *b* and *c* are real non-negative coefficients. After some straightforward algebra, it can be verified that $A_{1}$ has eigenvalues
(27)$$x_{k}\pm \sqrt{b^{2}-c^{2}}\sqrt{x_{k}}\;\mathrm{with}\;k=1,2,\ldots ,M$$
with eigenvectors
(28)$$|x_{k}\rangle \pm \sqrt{\frac{b-c}{b+c}}|y_{k}\rangle .$$
As the eigenvalues are real, if b≥c, and complex conjugate, if b<c, $A_{1}$ is pseudo-Hermitian Hamiltonian. Inspired by Ref. [1], we make the ansatz
(29)$$\mu =\exp\left(\beta \hat{S}\right)\exp\left(\alpha \hat{T}\right)$$
that replaced in Equation (Equation 28) allow us to derive, with the help of the above relations that
(30)$$\begin{array}{cc} h=\hat{U} & +\left[R\cosh\left(2\beta \sqrt{\hat{U}}\right)-\frac{i\hat{T}}{\sqrt{\hat{U}}}\sinh\left(2\beta \sqrt{\hat{U}}\right)\right]\left[b\cosh\left(2\alpha \hat{U}\right)+c\sinh\left(2\alpha \hat{U}\right)\right]\\ \; & +iS\left[b\sinh\left(2\alpha \hat{U}\right)+c\cosh\left(2\alpha \hat{U}\right)\right]-i\dot{\beta }\hat{S}\\ \; & +\left[\hat{R}\sinh\left(2\beta \sqrt{\hat{U}}\right)-\frac{i\hat{T}}{\sqrt{\hat{U}}}\cosh\left(2\beta \sqrt{\hat{U}}\right)\right]\sqrt{\hat{U}}\dot{\alpha }. \end{array}$$
By imposing that the parameters α and β satisfy the differential equations
(31)$$\dot{\alpha }=-\frac{\tanh(2\beta \sqrt{\hat{U}})}{\sqrt{\hat{U}}}\left[b\cosh\left(2\alpha \hat{U}\right)+c\sinh\left(2\alpha \hat{U}\right)\right]$$
and
(32)$$\dot{\beta }=b\sinh\left(2\alpha \hat{U}\right)+c\cosh\left(2\alpha \hat{U}\right),$$
the non-Hermitian terms are removed and the Hermitian matrix
(33)$$h=\hat{U}+\frac{b\cosh\left(2\alpha \hat{U}\right)+c\sinh\left(2\alpha \hat{U}\right)}{\cosh(2\beta \sqrt{\hat{U})}}\hat{R}=\hat{U}+\nu \left(t\right)\hat{R}.$$
is obtained.

Although these equations show great similarity to those of [1], an important difference lies in the fact that for them α and β are numbers while for us they are matrices that are functions of $\hat{U}$. Nevertheless, relying in the commutativity of *U* with the other matrices, we assume that *U* can be treated as a number, such that our equations can be solved following the same steps. Thus, by inverting (Equation 32) we obtain
(34)$$\tanh\left(2\alpha \hat{U}\right)=\frac{-bc+\dot{\beta }\sqrt{b^{2}-c^{2}+{\dot{\beta }}^{2}}}{b^{2}+{\dot{\beta }}^{2}}.$$
Next, to decouple Equations (Equation 31) and (Equation 32), we first combine them to get
(35)$$\dot{\alpha }=-\frac{1}{\sqrt{\hat{U}}}\tanh\left(2\beta \sqrt{\hat{U}}\right)\sqrt{b^{2}+{\dot{\beta }}^{2}-c^{2}},$$
then, by taking the derivative of (Equation 32), we obtain
(36)$$\ddot{\beta }=-2\sqrt{\hat{U}}\tanh\left(2\beta \sqrt{\hat{U}}\right)\left({\dot{\beta }}^{2}+b^{2}-c^{2}\right)$$
such that if a new variable $\sigma =\sinh(2\beta \sqrt{\hat{U}})$ is defined, it is found that it satisfies a harmonic oscillator equation that can be solved as
(37)$$\sinh\left(2\beta \sqrt{\hat{U}}\right)=\frac{C_{1}}{\sqrt{b^{2}-c^{2}}}\sin\left[2\sqrt{\hat{U}\left(b^{2}-c^{2}\right)}\left(t+C_{2}\right)\right].$$
Once, the dependence with time of the parameter β is obtained, we can replace it in Equation (Equation 35) to have
(38)$$\sqrt{\hat{U}}\dot{\alpha }=-\frac{\sqrt{\left(b^{2}-c^{2}\right)\left(C_{1}^{2}+b^{2}-c^{2}\right)}C_{1}\sin\left[2\sqrt{\hat{U}\left(b^{2}-c^{2}\right)}\left(t+C_{2}\right)\right]}{b^{2}-c^{2}+C_{1}^{2}\sin^{2}\left[2\sqrt{\hat{U}\left(b^{2}-c^{2}\right)}\left(t+C_{2}\right)\right]}$$
that can be solved as
(39)$$2\hat{U}\alpha =\mathrm{atanh}\left(\frac{C_{1}\cos\left[2\sqrt{\hat{U}\left(b^{2}-c^{2}\right)}\left(t+C_{2}\right)\right]}{\sqrt{C_{1}^{2}+b^{2}-c^{2}}}\right)+B,$$
where *B* is an integration constant which with the value B=−atanh(c/b) yields the same expression
(40)$$\exp\left(4\alpha \hat{U}\right)=\left(\frac{b-c}{b+c}\right)\frac{\sqrt{C_{1}^{2}+b^{2}-c^{2}}+C_{1}\cos\left[2\sqrt{\hat{U}\left(b^{2}-c^{2}\right)}\left(t+C_{2}\right)\right]}{\sqrt{C_{1}^{2}+b^{2}-c^{2}}-C_{1}\cos\left[2\sqrt{\hat{U}\left(b^{2}-c^{2}\right)}\left(t+C_{2}\right)\right]},$$
in [1] (we remark that the above value of the integration constant is necessary to Equation (Equation 34) be satisfied). From it, we obtain
(41)$$\nu \left(t\right)=\frac{\left(b^{2}-c^{2}\right)\sqrt{C_{1}^{2}+b^{2}-c^{2}}}{C_{1}^{2}+b^{2}-c^{2}-C_{1}^{2}\cos^{2}\left[2\sqrt{\hat{U}\left(b^{2}-c^{2}\right)}\left(t+C_{2}\right)\right]}.$$
and
(42)$$\nu_{I}=\int^{t}\nu \left(t^{\prime }\right)dt^{\prime }=\frac{1}{2\sqrt{U}}\arctan\left(\frac{\sqrt{C_{1}^{2}+b^{2}-c^{2}}}{\sqrt{b^{2}-c^{2}}}\tan\left[2\sqrt{\hat{U}\left(b^{2}-c^{2}\right)}\left(t+C_{2}\right)\right]\right)=\frac{\gamma (\hat{U})}{\sqrt{\hat{U}}}.$$
such that we now have the unitary operator $\exp\left(-i\hat{U}t-i\gamma \left(\hat{U}\right)\frac{\hat{R}}{\sqrt{\hat{U}}}\right)$ to make the time evolution of initial states.

Let us start evolving just the single state given by Equation (Equation 24). Applying the term $\exp\left(-i\hat{U}\right)$ it just produces an overall phase and, for the other term, we have
(43)$$\gamma \left(\hat{U}\right)\frac{\hat{R}}{\sqrt{\hat{U}}}=\sum_{k=1}^{M}|x_{m}\rangle \gamma \left(x_{m}\right)\langle y_{k}|+\sum_{l=1}^{M}|y_{l}\rangle \gamma \left(y_{l}\right)\langle x_{l}|$$
such that after some algebra, we obtain
(44)$$e^{-i\hat{U}-i\frac{\gamma (\hat{U})R}{\sqrt{\hat{U}}}}\left(\cos\frac{\theta }{2}|x_{k}\rangle +e^{i\phi }\sin\frac{\theta }{2}|y_{k}\rangle \right)=e^{-ix_{k}t}\left[\cos\left(\frac{\theta }{2}-\gamma_{k}\right)|x_{k}\rangle +e^{i\phi }\sin\left(\frac{\theta }{2}-\gamma_{k}\right)|y_{k}\rangle \right].$$
The above equation shows that the generator produces a rotation and the evolved density matrix is then
(45)$$\begin{array}{cc} \rho = & \cos^{2}\left(\frac{\theta }{2}-\gamma_{k}\right)|x_{k}\rangle \langle x_{k}|+\sin^{2}\left(\frac{\theta }{2}-\gamma_{k}\right)|y_{k}\rangle \langle y_{k}|\\ \; & +\cos\left(\frac{\theta }{2}-\gamma_{k}\right)\sin\left(\frac{\theta }{2}-\gamma_{k}\right)\left(e^{i\phi }|x_{k}\rangle \langle y_{k}|-e^{-\phi }|y_{k}\rangle \langle y_{k}|\right), \end{array}$$
which is a pure state. If the partial trace is taken or, in physical terms, if the interference terms are removed, the states become mixed in which $\cos^{2}\left(\frac{\theta }{2}-\gamma_{k}\right)$ and $\sin^{2}\left(\frac{\theta }{2}-\gamma_{k}\right)$ are the probabilities of finding, in a measurement, the system to be in PHQHP or in QHPHQ parts, respectively. In this case, if b>c, the probabilities oscillate, while for b<c, asymptotically, when *t* goes to infinity, $\gamma_{k}$ goes to $\gamma_{\infty }=\frac{1}{2}\arctan\sqrt{\frac{C_{1}^{2}+b^{2}-c^{2}}{c^{2}-b^{2}}}$ and the probabilities reach fixed values.

Turning now to entanglement, one would expect to be natural to study the bipartite division PHQHP and QHPHQ, but, taking into account that the eigenstates of our matrices are qubits, we work with them instead. Considering the present case of the generator $\hat{R},$ Equation (Equation 19), we use pairs (m,n) of its eigenstates to construct the Bell states
(46)$$|\Phi_{R}^{\pm }\left(0\right)\rangle =\frac{1}{\sqrt{2}}\left(|X_{m}^{+}\rangle |X_{n}^{+}\rangle \pm |X_{m}^{-}\rangle |X_{n}^{-}\rangle \right),$$
with which an initial state
(47)$$|\chi (0)\rangle =\cos\frac{\theta }{2}|\Phi_{R}^{+}\left(0\right)\rangle +\sin\frac{\theta }{2}|\Phi_{R}^{-}\left(0\right)\rangle ,$$
is defined. In terms of the qubit states, this initial state can be spanned as
(48)$$|\chi (0)\rangle =\Lambda_{11}|X_{m}^{+}\rangle |X_{n}^{+}\rangle +\Lambda_{12}|X_{m}^{+}\rangle |X_{n}^{-}\rangle +\Lambda_{21}|X_{m}^{-}\rangle |X_{n}^{+}\rangle +\Lambda_{22}|X_{m}^{-}\rangle |X_{n}^{-}\rangle$$
and a density matrix $\rho_{ij;kl}=\Lambda_{ij}\Lambda_{kl}$ where
(49)$$\Lambda \left(0\right)=\frac{1}{\sqrt{2}}\left(\begin{array}{cc} \cos\frac{\theta }{2}+\sin\frac{\theta }{2} & 0\\ 0 & \cos\frac{\theta }{2}-\sin\frac{\theta }{2} \end{array}\right)$$
follows. Then, by taking partial trace, the *n*-states are removed and the reduced matrix density
(50)$$\rho_{ik}^{m}\left(0\right)=\sum_{l=1}^{2}\Lambda_{il}\Lambda_{kl}=\frac{1}{2}\left(\begin{array}{cc} 1+\sin\theta & 0\\ 0 & 1-\sin\theta \end{array}\right)$$
is obtained. We observe that for θ equal to zero or π, as the initial state turns out to be one of the Bell states, the entanglement is maximum. On the other hand, for $\theta =\frac{\pi }{2},$ the initial state is a pure state.

Evolving now the above initial state, we observe that as
(51)$$\frac{\gamma \left(\hat{U}\right)\hat{R}}{\sqrt{\hat{U}}}|\Phi_{R}^{\pm }\left(0\right)\rangle =\left(\gamma_{m}+\gamma_{n}\right)|\Phi_{R}^{\mp }\left(0\right)\rangle ,$$
it is deduced that
(52)$$e^{-i\frac{\gamma (\hat{U})R}{\sqrt{\hat{U}}}}|\Phi_{R}^{\pm }\rangle =|\Phi_{mn}^{\pm }\left(t\right)\rangle =\cos\Delta |\Phi_{R}^{\pm }\left(0\right)\rangle -i\sin\Delta |\Phi_{R}^{\mp }\left(0\right)\rangle ,$$
where $\Delta =\gamma_{m}+\gamma_{n}.$ From this result, it follows that
(53)$$|\chi (t)\rangle =\left(\cos\frac{\theta }{2}\cos\Delta -i\sin\frac{\theta }{2}\sin\Delta \right)|\Phi_{R}^{+}\left(t\right)\rangle +\left(\sin\frac{\theta }{2}\cos\Delta -i\cos\frac{\theta }{2}\sin\Delta \right)|\Phi_{R}^{-}\left(t\right)\rangle ,$$
or, in terms of the eigenstates,
(54)$$\begin{array}{c} |\chi (t)\rangle =\frac{1}{\sqrt{2}}\left[\left(\cos\frac{\theta }{2}+\sin\frac{\theta }{2}\right)\cos\Delta -i\left(\sin\frac{\theta }{2}+\cos\frac{\theta }{2}\right)\sin\Delta \right]|X_{m}^{+}\rangle |X_{n}^{+}\rangle \\ +\frac{1}{\sqrt{2}}\left[\left(\cos\frac{\theta }{2}-\sin\frac{\theta }{2}\right)\cos\Delta -i\left(\sin\frac{\theta }{2}-\cos\frac{\theta }{2}\right)\sin\Delta \right]|X_{m}^{-}\rangle |X_{n}^{-}\rangle . \end{array}$$
Taking then the partial trace, the time-dependent reduced density matrix
(55)$$\rho_{ik}^{m}\left(t\right)=\frac{1}{2}\left(\begin{array}{cc} 1+\sin\theta \cos2\Delta & 0\\ 0 & 1-\sin\theta \cos2\Delta \end{array}\right)$$
is obtained and the von Neumann entropy
(56)$$S=-\lambda_{1}\log\lambda_{1}-\lambda_{2}\log\lambda_{2}$$
is calculated with
(57)$$\lambda_{1}=\frac{1}{2}\left(1+\sin\theta \cos2\Delta \right)$$
and
(58)$$\lambda_{2}=\frac{1}{2}\left(1-\sin\theta \cos2\Delta \right).$$
We remark that the entangled qubits are chiral states of the matrix *R* and of the pseudo-Hermitian matrix $A_{1}.$ Their entanglement is done in a ++ and −− way such that if the state *m* is in the positive (negative) state then the *n* is in the positive (negative).

In Figure 1 and Figure 2, it is shown the time evolution of the von Neumann entropy in the regimes of real, b>c, and complex conjugate eigenvalues, b<c, respectively. In both cases, N=6 and M=2 such that there are only the ground state and the first excited state; moreover, as $\theta =\frac{\pi }{2}$, the initial is a pure state. In Figure 1, the calculation is performed with two sample matrices and the effect of the randomness of the eigenvalues is exhibited. This randomness is also present in the fact that the oscillations do not show a constant period. As a matter of fact, for larger matrices, that is N≫M the randomness is suppressed.

In Figure 2, the “eternal life of the entropy” effect, as the authors of Ref. [1] called it, is shown. Starting from its value at the pure state, the entropy reaches the value of maximum entanglement and then decays to a constant value.

### 3.2. Entanglement of Bipartite States

We now consider the matrix
(59)$$A_{2}=\hat{U}+b\frac{\hat{T}}{\sqrt{\hat{U}}}-ic\hat{S},$$
in which as before, *b* and *c* real non-negative coefficients. It can be again verified that $A_{2}$ has eigenvalues
(60)$$x_{k}\pm \sqrt{b^{2}-c^{2}}\sqrt{x_{k}}\;\mathrm{with}\;k=1,2,\ldots ,M$$
with unnormalized eigenvectors
(61)$$\left(b\pm \sqrt{b^{2}-c^{2}}\right)|x_{k}\rangle +c|y_{k}\rangle .$$
The eigenvalues are real, if b≥c, and complex conjugate, if b<c, $A_{2}$ is pseudo-Hermitian with respect to the metric P−Q.

Replacing the ansatz
(62)$$\mu =\exp\left(\zeta \hat{S}\right)\exp\left(-\delta \hat{T}\right)$$
in Equation (Equation 28) we find that if the time-dependent parameters ζ and δ satisfy the differential equations
(63)$$\dot{\delta }=-\frac{\tanh(2\zeta \sqrt{U})}{\sqrt{\hat{U}}}\left[b\cosh\left(2\delta \sqrt{\hat{U}}\right)+c\sinh\left(2\delta \sqrt{\hat{U}}\right)\right]$$
and
(64)$$\dot{\zeta }=b\sinh\left(2\delta \sqrt{\hat{U}}\right)+c\cosh\left(2\delta \sqrt{U}\right),$$
then, following the same steps employed in the $A_{1}$ Hamiltonian, the non-Hermitian terms are removed and the Hermitian matrix
(65)$$h=\hat{U}+\frac{b\cosh\left(2\delta \sqrt{\hat{U}}\right)+c\sinh\left(2\delta \sqrt{\hat{U}}\right)}{\cosh(2\zeta \sqrt{\hat{U})}}\hat{R}=\hat{U}+\xi \left(t\right)\frac{T}{\sqrt{U}}.$$
is obtained.

Evolving first just the single state given by Equation (Equation 24) with $\xi_{I}\left(t\right)$ equal to $\nu_{I}\left(t\right)$, Equation (Equation 42), we find the evolved state
(66)$$\begin{array}{cc} |\chi (t)\rangle = & \exp\left(-iU-i\gamma \frac{\hat{T}}{\hat{U}}\right)\left(\cos\frac{\theta }{2}|x_{k}\rangle +\sin\frac{\theta }{2}e^{i\phi }|y_{k}\rangle \right)\\ = & \exp\left(-ix_{k}\right)\left(e^{-i\gamma_{k}}\cos\frac{\theta }{2}|x_{k}\rangle +e^{i\gamma_{k}+i\phi }\sin\frac{\theta }{2}|y_{k}\rangle \right). \end{array}$$
and the evolved density matrix
(67)$$\begin{array}{cc} \rho =|\chi (t)\rangle \langle \chi (t)|= & \cos^{2}\frac{\theta }{2}|x_{k}\rangle \langle x_{k}|+\sin^{2}\frac{\theta }{2}|y_{k}\rangle \langle y_{k}|\\ \; & +\cos\frac{\theta }{2}\sin\frac{\theta }{2}\left(e^{-2i\gamma_{k}-i\phi }|x_{k}\rangle \langle y|+e^{2i\gamma_{k}+i\phi }|y_{k}\rangle \langle y_{k}|\right), \end{array}$$
which is a pure state. The evolution produces a phase and, consequently, the quantum measurement is not modified.

Turning now to entanglement, we use the eigenstates of the operator $\hat{T},$ Equation (Equation 19), to constructing the Bell states
(68)$$|\Phi_{T}^{\pm }\left(0\right)\rangle =\frac{1}{\sqrt{2}}\left(|x_{m}\rangle |x_{n}\rangle \pm |y_{m}\rangle |y_{n}\rangle \right)$$
and define the initial state
(69)$$|\chi (0)\rangle =\cos\frac{\theta }{2}|\Phi_{T}^{+}\left(0\right)\rangle +\sin\frac{\theta }{2}|\Phi_{T}^{-}\left(0\right)\rangle .$$
Using alternatively tensor product we have
(70)$$|\Phi_{T}^{\pm }\left(t\right)\rangle =e^{-i\gamma \left(\hat{U}\right)\frac{\hat{T}}{\hat{U}}}⊗e^{-i\gamma \left(\hat{U}\right)\frac{\hat{T}}{\hat{U}}}|\Phi_{T}^{\pm }\left(0\right)\rangle =\cos\Delta |\Phi_{T}^{\pm }\left(0\right)\rangle -i\sin\Delta |\Phi_{T}^{\mp }\left(0\right)\rangle ,$$
and, after taking the partial trace, the same time-dependent reduced density matrix
(71)$$\rho_{ik}^{m}\left(t\right)=\frac{1}{2}\left(\begin{array}{cc} 1+\sin\theta \cos2\Delta & 0\\ 0 & 1-\sin\theta \cos2\Delta \end{array}\right).$$
is obtained. We have, therefore, ended up with the same von Neumann entropy of the previous case, but with a different underlying physics. Here the qubits are the eigenstates of the matrix *T* and also of the pseudo-Hermitian $A_{2}.$ The entanglement here is such that the positive and the negative values correspond to the bipartite PHQHP and QHPHQ parts of the system. If the *m*-qubit is +, both states are in the first part while if it is −, both are in second part.

## 4. Conclusions

The transition from an oscillatory regime, Figure 1, to a regime in which the oscillations are damped, Figure 2, is, of course, a consequence of moving from real to complex conjugate eigenvalues. The new feature, however, is that when eigenvalues are complex, as they are conjugate, although the system is open there is a balance between gains and losses that leads, in the time evolution, to an asymptotically stationary situation. This is a generic aspect of the evolution of *PT*-symmetric or, in more general terms, of pseudo-Hermitian Hamiltonians as recent investigations of squeezed states show [33,34]. With respect to entanglement, as to avoid decoherence is a necessary constraint in any quantum computer project, the periodic vanishing of the entropy, the so-called “sudden death” [35,36] makes the real eigenvalues regime unpractical. This explains the importance of the “eternal life of the entropy” effect present in the time evolution of the complex conjugate eigenvalue regime. The fact that we were able to construct an abstract random matrix model suggests the universality of the effect. Moreover, as the model ended up being made of objects, i.e., Pauli-like matrices, qubits, Bell states, which are at the core of quantum information theory and experiment, makes us believe that our work enhances the potential practical application of the effect.

## Figures and Tables

**Figure 1 entropy-22-01109-f001:**
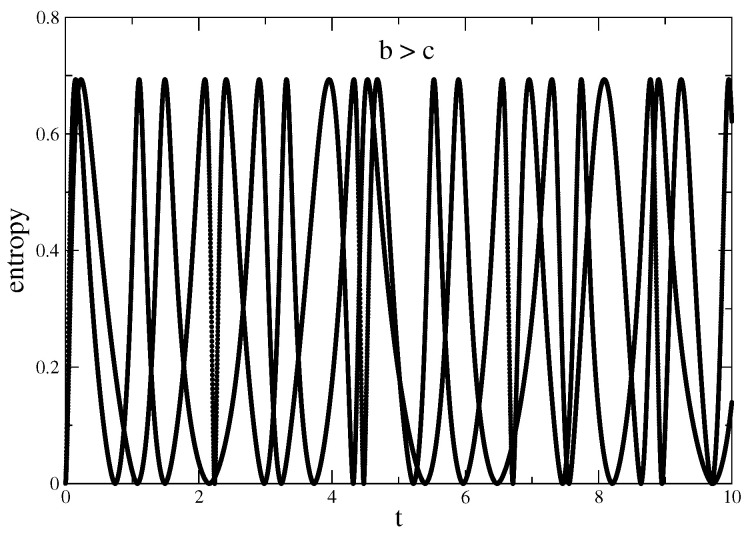
With $\theta =\frac{\pi }{2}$, von Neumann entropy in function of time for two random matrices with the values $C_{1}=2,$b=1.2 and c=1.

**Figure 2 entropy-22-01109-f002:**
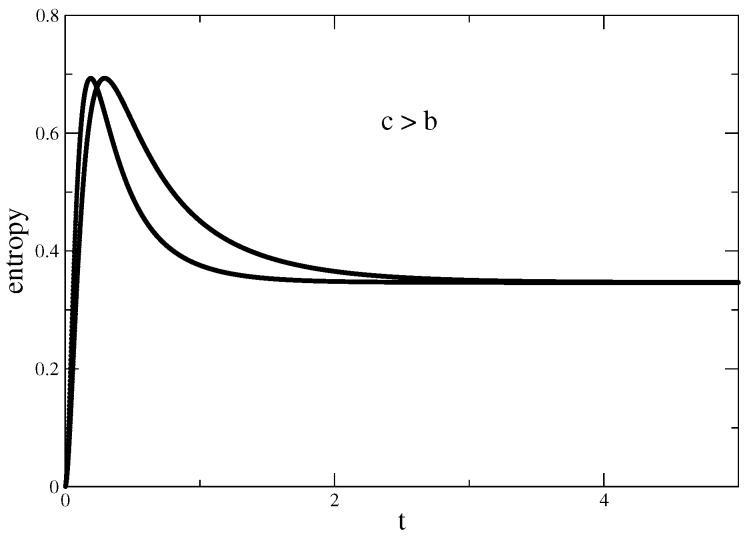
With $\theta =\frac{\pi }{2}$, von Neumann entropy in function of time for two random matrices with the values $C_{1}=2,$b=1.0 and c=1.2.

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
