# Peer review of "Entanglement of Pseudo-Hermitian Random States"

_entropy, 2020, doi:10.3390/e22101109_

Round 1

Reviewer 1 Report

    The authors have developed an interesting algebraic scheme for the analysis of PT - symmetric Hamiltonians and apply this scheme for studying entanglement properties. In general the manuscript leaves me with a positive impression. This being said, I consider it too long and containing a large amount of quite secondary and technical information (especially Sec.2.1), which makes the manuscript boring to read. In addition, in the Conclusions section, the authors mostly give a short description of the manuscript, instead of pointing out on the physical aspects of the problem
    I would recommend an amended version of this manuscript for publication, but not in the present form.

Author Response

We really thank the referee for his (her) report. Taking into account the suggestions

and the critics we have done the following:

  1. To make the article more readable, the second section was split in two sections, one in which the matrices of the model are introduced and their properties are discussed and another one in which the pseudo-Hermitian Hamiltonians are introduced, their time evolutions are derived and the entanglement of their states are obtained. A redundant paragraph was also removed. Besides, at the end of the introduction, a guide to the structure of the article was added.
  2. Turning now to the most important point correctly raised by he (she), to fix it the conclusion section was entirely rewritten in order to address "the physical aspects of the problem". We hope that this will be enough.

Reviewer 2 Report

I have carefully read the manuscript entropy-939961. The Authors
study an abstract model of pseudo hermitian random matrices to
study the entanglement of the states of a non-hermitian bosnic
system. They applied their formalism, which includes the
construction of a proper metric, to two different
Hamiltonians (Eqs. (29) and (62). The Authors analyse the
evolution in time of chiral and bipartite states, respectively.
They found that in the regimen of broken symmetry the initial
state evolves to a steady state with constant entropy, as it was
previously reported in their Ref. (1). I would like to refer the
Authors to some previous works that also show that in the
broken regimen the initial state evolves to a non-trivial
steady state, i.e.: Journal of Mathematical
Physics 60(1)(2019)012106, Phys. Scripta 94 (2019) 85220. In my opinion, the manuscript makes an original contribution to
the study of non-hermitian quantum dynamics. From my previous observations, I recommend the manuscript for
publication in Entropy after the previous suggestion have been
addressed.

Author Response

We thank the referee for recommending the publication of our article and also to call our attention to the two important publications related to our work that we were not aware of. We have already included them in the references.

Reviewer 3 Report

Dear Editor and Authors,

This is a report on manuscript Entropy-939961, “Entanglement of pseudo-Hermitian random states,” that follows a recent manuscript showing so-called “eternal life of entropy” [PRA 100, 101102 (2019)] and extends that formalism to pseudo-Hermitian random matrices.

The manuscript shows results that seem correct using what seems a graded Lie algebra approach from a random matrix point of view. However, I find the introduction lacking in the sense of providing context or motivation beyond the link to [PRA 100, 101102 (2019)]. Furthermore, the manuscript requires copy-editing by a native speaker to improve the flow; e.g., the first sentence in the abstract.

I am positive about recommending this manuscript for publication once these and the following comments are addressed,

  1. The introduction needs improvement in providing a coherent narrative that gives the reader context, motivation and a roadmap to the rest of the manuscript.
  2. How can we relate all the mathematical implications of similarity transformations, new metrics for the internal product, the use of random ensembles, etc to experimental or theoretical situations in the literature?
  3. How is this model analogous to that of quantum gates?
  4. In the introduction, please, provide a paragraph that informs the reader about the rest of the manuscript, why it is presented in this order, how the section interconnect with each other, etc.
  5. The model. Why is this model relevant? Does it relate to experiments or theory? Please inform the reader why this bipartite separation is chosen.
  6. The structure of the Hermitian traceless matrices seems pretty similar to a graded Lie algebra.
  7. Is the structure of R, S, and T that of a Pauli Matrix or that of a nilpotent charge generator of a graded Lie algebra? In SUSY QM, Pauli j=1/2 matrices are at the core of these structures but have other symmetry on top of them.
  8. Why is a density matrix needed here? The treatment is for a closed system, all results presented are for pure states. In Ref. [1], the system is a bipartite system where one block is considered as the environment, thus, results on the dynamics of the second block require density matrices as they are just part of the whole system. Here the density matrix $\rho_{A}$ lives in the whole space of the system as $i=1, \ldots, N$.
  9. Eq. 32 is an ansatz based on Wei-Norman decomposition due to the structure of the graded algebra. This should be explicit to the reader. Furthermore, do the results following this equation arise due to the similitudes of the underlying symmetry to a deformed complexified su(2) or finite size su(1,1) algebra?

In general, I find the manuscript technically extensive and believe that it will benefit greatly from a coherent and concise narrative that explicitly connects the technical results with physical ideas or systems.

Round 2

Reviewer 3 Report

Dear Editor and Authors, 

The authors answered my concerns to the best of their abilities. 

I still find the manuscript lacking regarding narrative and motivation but the author's answer suggest this will not change further down the road.

The results have archival value and could be published in Entropy if so the editor decides.